# Pain Assessment for Patients with Dementia and Communication Impairment: Feasibility Study of the Usage of Artificial Intelligence-Enabled Wearables

**DOI:** 10.3390/s24196298

**Published:** 2024-09-29

**Authors:** Mehdi Snene, Christophe Graf, Petra Vayne-Bossert, Sophie Pautex

**Affiliations:** 1Geneva School of Economics and Management, University of Geneva, 1200 Geneva, Switzerland; 2United Nations Secretary General’s Envoy on Technology, New York, NY 10017, USA; 3Department of Readaptation and Geriatrics, Geneva University Hospitals, 1200 Geneva, Switzerland; christophe.graf@hcuge.ch (C.G.); petra.vayne-bossert@hcuge.ch (P.V.-B.); 4Department of Readaptation and Geriatrics, University of Geneva, 1200 Geneva, Switzerland

**Keywords:** wearables, AI, pain, digital biomarkers, real time, dementia

## Abstract

Background: Recent studies on machine learning have shown the potential to provide new methods with which to assess pain through the measurement of signals associated with physiologic responses to pain detected by wearables. We conducted a prospective pilot study to evaluate the real-world feasibility of using an AI-enabled wearable system for pain assessment with elderly patients with dementia and impaired communication. Methods: Sensor data were collected from the wearables, as well as observational data-based conventional everyday interventions. We measured the adherence, completeness, and quality of the collected data. Thereafter, we evaluated the most appropriate classification model for assessing the detectability and predictability of pain. Results: A total of 18 patients completed the trial period, and 10 of them had complete sensor and observational datasets. We extracted 206 matched records containing a 180 min long data segment from the sensor’s dataset. The final dataset comprised 153 subsets labelled as moderate pain and 53 labelled as severe pain. After noise reduction, we compared the recall and precision performances of 14 common classification algorithms. The light gradient-boosting machine (LGBM) classifier presented optimal values for both performances. Conclusions: Our findings tended to show that electrodermal activity (EDA), skin temperature, and mobility data are the most appropriate for pain detection.

## 1. Introduction

In recent years, patient-reported outcomes (PROs) have gained popularity. Patients’ perceptions and evaluations of symptom management or even overall quality of life have attracted considerable research attention. Furthermore, the regular collection of PRO data can increase the quality of clinical care through the early identification of clinical problems, improve patient–clinician communication, enhance clinicians’ awareness of symptoms, support symptom management, preserve quality of life, and even increase overall survival [1,2,3]. Most of these PRO collections involve personal resources (interviews, calls, emails, etc.), electronic technologies, and, most importantly, significant patient participation.

All these modes use self-reports and open-ended questions to elicit information related to the qualitative and quantitative outcomes of patients [4,5].

Pain is a recurrent symptom explored in different PRO scales, and is negatively correlated with quality of life. For patients with communication abilities, pain is always evaluated through self-reports and mediated by a patient’s subjective perceptions [6,7]. Dementia causes serious and permanent barriers to pain assessment through self-reports. All of the pain assessment tools for patients with dementia and communication impairment rely on appraisals by another person; thus, PROMs on pain are often difficult to obtain in dementia, and alternative measurement methods are required [8].

Physiological signals such as heart rate or blood pressure can therefore provide important information, especially in the assessment of non-communicating patients with dementia. Although pain assessment methods implemented by multi-modality signals have been confirmed to be highly effective, with some even outperforming single-signal modes considerably, research on the exploitation of these physiological signals measured with wearables is scarce, owing to their limited validity and practical limitations [9,10]. Furthermore, there are insufficient studies based on automatic measurements of physiological signals for pain in subject groups with mild or severe cognitive impairment [10]. We aim to determine the feasibility of the continuous monitoring of patients with dementia and communication impairment using wearables and machine learning. To this end, we have conducted our study in a real-world setting without using provoked pain stimuli.

## 2. Materials and Methods

### 2.1. Study Design and Participants

This is a feasibility pilot study. Participants were recruited from the Internal Medicine and Rehabilitation Division of the University Hospitals of Geneva. As it was a feasibility study, we aimed to recruit approximately 20 patients with different degrees of severity of dementia. The inclusion criteria were in-patients aged over 18 years old with a diagnosis of dementia of different origins (Alzheimer’s disease, vascular or mixed dementia) and experiencing pain according to a proxy or self-assessment, with an expected length of in-hospital stay of more than two weeks. Consent to participate in the study was obtained from the patient themself when possible. In the absence of the capacity of discernment, proxy consent was obtained.

The exclusion criteria were patients with severe behavioural disorders, which did not allow them to be equipped with the wearable, defined as having a Cohen–Mansfield agitation inventory > 160, as well as patients identified as being in their last days of life. Patients who could not consent to the study and did not have a family member acting as a caregiver who could have given consent for a patient were also excluded.

### 2.2. Data Collection

Before starting the collection of physiological data via the wearable, the following dataset was collected:Characteristics of each patient: age; type and severity of dementia; mini-mental status examination [11]; comorbidities (Charlson comorbidity score); pain origin (osteoarticular; inflammatory; neurogenic; visceral; and other); and prescribed regular analgesics as well as breakthrough pain medications.Quality of sleep (0 to 10) was assessed in the morning each day, by asking the patient.Pain assessment: At least once a day through self-assessment with a numerical rating scale (rated from 0 to 10) if the patients were able to communicate, and additionally at any time a patient complained of pain [12]. Otherwise, at least once a day with an Algoplus scale (rated from 0 to 5) [13], completed by a nurse for patients with verbal communication difficulties [14]. All pain scores were documented and collected from the medical charts.Vital signs were collected daily, including blood pulse (number per minute), blood pressure (mmHg/mmHg), temperature (number °C), and respiratory rate (number per minute).

## 3. Wearable

The Empatica E4, a commercially available wristband, was used to monitor the data. The participants wore the wristband to monitor their physiological signals continuously, i.e., 24 h/24, for fourteen days. The E4 contains four primary sensors: photoplethysmography (PPG, rate of 64 Hz) to provide the BVP (blood variation pressure), from which the HR (heart rate), HRV (heart rate variation), and IBI (interbeat interval) are derived; tonic electrodermal activity (EDA, rate of 4 Hz), which reports the electrical properties of the skin; a three-axis accelerometer (rate of 32 Hz), to capture motion-based activity; and an infrared thermopile (rate of 4 Hz) to measure peripheral skin temperature [15]. Physiological data from these sensors were stored in the on-board memory of the E4, and were downloaded daily by the research staff. Owing to its limited battery and storage capacity, two E4 wristbands were allocated for each patient. There was no interruption in signal measurement when the device was running out of battery because it was immediately replaced by a fully charged one. The study staff were instructed about the proper handling and wearing of the E4 wristband. The staff oversaw the wristband replacement daily and the transfer of data to the E4 manager tool.

### 3.1. Data Pre-Processing

Several pre-processing steps were performed on our dataset to prepare it for analysis. The first step included removing incomplete observation days, where pain measurements were missing. In this case, this step dismissed daily data that were not linked to at least one daily pain assessment. The second step was to remove data from patients that did not complete at least one week of the clinical trials. At the end of this processing step, only nine patients’ data were used for the analysis.

Raw sensor signals retrieved from wearable sensors were collected in different frequencies. The {Iv1, Iv2, Iv3} set composed data from discrete time series sampled at 64 Hz; the set of {Iv4} was sampled at 4 Hz, while the {Iv5, Iv6, Iv7} set was sampled at 32 Hz. For consistency, the highest frequencies were downsampled to the same frequency of the lowest frequency data (4 Hz). This resampling was achieved by averaging in blocks of the frequency (64 Hz: blocks of 16; 32 Hz: blocks of 8).

Based on the assumption that a patient’s pain score does not usually change quickly within a short period, we extracted a 180 min data segment centred on the assessment time of the pain score (i.e., 120 and 60 min before and after the recorded time, respectively). For example, if a pain score was recorded at 10:00 p.m., then it was matched with the time series recorded from 08:00 p.m. to 11:00 p.m. In addition to the daily regular pain assessment, caregivers were instructed to conduct a new measurement if patients expressed abnormal behaviours or showed signs that indicated a change in pain intensity. For the intervention, we assumed that the time between the appearance of the signs to their notification by caregivers and then documentation in the medical chart by the nursing staff, followed by proceeding to a new pain assessment was, on average, 120 min, considering the existing resources and time availability. If there was any pain relief intervention (e.g., analgesic), 60 min was considered as the standard average time for the intervention to be effective in pain reduction.

### 3.2. Ethical Approval

This study was approved by the ethics committee of “The Republic and Canton of Geneva”, number 2018-01835. All subjects or legal representatives were fully informed about the procedures, risks, and benefits of the study. In addition, written informed consent was obtained from all of the subjects or from the surrogate representative before their inclusion in the study.

### 3.3. Data Analysis and Modelling

There are different methods in the literature for determining pain and/or pain levels, including support vector machine (SVM), logistic regression, decision tree, and random forest [16]. Each of these methods has proven to be effective in specific settings for pain or stress measurement. The technical characteristics of each machine learning method can be identified in terms of the sensing devices (e.g., accelerometer, GSR, and HRV), mathematical models (e.g., artificial neural networks, random forest), and algorithms used to determine the optimum solution (e.g., stochastic gradient descent, Bayesian variational inference). We can distinguish two widely used categories for the prediction of pain or a pain score: a regression or a classification approach. In our feasibility study, we included three different classes, as there were three distinct possible pain scores (0, 1, and 2) and eight numeric input variables of varying scales.

We implemented a comparison between 14 widely used classification algorithms: LGBM (light gradient-boosting machine), RF (random forest), ET (extra tree classifier), DT (decision tree), GBC (gradient-boosting classifier), KNN (K neighbours classifier), ADA (AdaBoost classifier), LDA (linear discriminant analysis), LR (logistic regression), RIDGE (ridge classifier), QDA (quadratic discriminant analysis), NB (naïve Bayes), SVM (SVM linear kernel), and DC (dummy classifier). Each algorithm was evaluated using 10-fold cross-validation, with the same random seed to ensure that the training data were split equally between each algorithm and evaluated in the same manner across the algorithms. For each of these classifiers, we calculated four comparison points represented by their average: accuracy, AUC, recall, and precision.

### 3.4. Outcome

Feasibility was evaluated by measuring the recruitment rate, dropout rate, a participant’s adherence to the assessment intervention, and reliable 24 h measurement. A secondary objective was to evaluate the feasibility and the efficiency of the used methodology for data collection to build, first, a labelled dataset annotated with analogue observations for pain intensity as assessed by the usual scales (NRS/Algoplus) and their matching physiological signals, and, second, a multivariate machine learning model that learns from the obtained dataset. We classified clinical pain states and predicted pain intensity in real time via uniquely using physiological signals from wearable devices.

### 3.5. Pilot Results—Statistics

The primary outcomes were pain, physiological signals, and observational measurements. For the analysis of the output, clinical characteristics and outcome measures were presented by the mean (standard deviation (SD)) or median (interquartile range (IQR)).

Concerning pain assessment, the numerical assessment scale (NRS) and Algoplus scores have been converted into a binned descriptive score (Table 1).

The same technique was also applied to the quality of sleep. Two scales were used: the first was scaled from 0 to 10, such that 10 was excellent quality and 0 was very bad, and the second was Boolean, with 0 as very poor quality and 1 as excellent. Both scaled to a single binned descriptive score.

The final dataset was composed from the following independent and dependant variables:Independent variables (symbol Iv): Iv1: heart rate (HR); Iv2: blood variation pressure (BVP); Iv3: interbeat interval (IBI); Iv4: electrodermal activity (EDA); Iv5: Acceleration_1 (ACC_1); Iv6: Acceleration_2 (ACC_2); and Iv7: Acceleration_3 (ACC_3);Dependent variables (symbol Dv): Dv1: pain (P); Dv2: sleep quality (SQ); and Dv3: pulse.

We operated a segmentation and selection of time intervals on the datasets. From the analogic observations for pain measurement, we selected 120 min prior to the time of the observation and 60 min after it.

## 4. Results

### 4.1. Participant Recruitment

From December 2019 to June 2021, 20 subjects (20 males; mean age of 84 (range of 64–91) years (standard deviation [SD] = 7.1)) were recruited in this study. Patients were monitored for an average of 312 (SD 12.4) hours, with an average of 1,198,000 (SD: 102,000) instances of raw data collected with the wearables and 364 objective data values per patient collected during the medical routines.

A total of 18 patients completed the trial period (Table 2). The mean MMSE score was 19.5 ± 30. Five patients did not have the ability to communicate. The average Charlson index was 7.6 (SD: 3.06). Fifteen patients declared an osteoarticular origin of pain, five a neurogenic origin, three an inflammatory origin, two a visceral origin, and two declared other origins of pain. Almost all patients (17/18) were receiving a daily regular analgesic. The average pain scores collected (average of the pain scores during the 14 days) while they were being monitored decreased in 7 out of 18 patients (38%). The pain score increased in two patients (11%) and remained stable in one patient (5%). There were incomplete data (44%) from eight patients.

### 4.2. Participants’ Adherence to the Assessment Intervention, Reliable 24 h Measurement

Two hundred and six matched records containing an 180 min long data segment and a pain score were collected. The dataset included 153 subsets labelled as moderate pain (score 1) and 53 labelled as severe pain (score 2). Every dataset comprised 10,800 records/Ivs. In total, 2,224,800 records (total size for all, “1 to 7”, was 15,573,600 instances of raw data) for each Iv distributed between moderate and severe pain episodes were collected. Considering the final data size, it was difficult to process the raw data directly in any analytical task. Consequently, we proceeded to feature extraction to transform a dataset in a data representation format that could be analysed.

To reduce random noise, an average filter was applied to the raw sensor signals, followed by the extraction of seven statistical features for each signal (as described in Table 3). By extracting features from the original raw signals, we reduced the size of the datasets while maintaining the properties of the original signal.

The number of samples for the three pain levels was 7975, 1444, and 442, indicating a high class imbalance among the three classes. Owing to the unequal distribution of datasets versus classes, we employed specific metrics for imbalanced data (Table 4). Imbalanced data refers to a classification problem where a class has an unequal number of observations that may result in introducing a bias in the prediction model towards the more common class. Accuracy is the total number of correct predictions divided by the total number of predictions made for a dataset; thus, it is inappropriate for imbalanced classification problems.

Two metrics were used for the comparison: the recall that measures how many positive class predictions compose all of the positive examples, and the precision that reflects the number of positive class predictions that are in the positive category. In this dataset, recall was the ratio of the number of correctly identified entities with a pain score over the total number of entities with the same pain score, whereas precision was the ratio of correctly identified entities with a pain score to the number of entities predicted by the model with the same pain score. Based on our dataset, the classifier that demonstrated optimal values for both metrics was the LGBM. A tuned 10-fold cross-validation result is summarised in Table 5.

In Figure 1, the performance of the selected classifiers is affected by the class imbalance problem. This is because the recall scores, precision scores, F1 scores, and support scores for no pain and mild pain are significantly higher than those for severe pain.

Employing all signal combinations can degrade the performance of the classification model, owing to feature redundancy. Some of the signals may be irrelevant, which impacts the classification result. Among the eight used signals, to choose a set of relevant signals, each feature must be computed for its discriminating ability when combined. To identify the most effective combination of signals, we adopted the SHapley Additive exPlanations (SHAP) value. SHAP values can remarkably explain specific predictions from a model by quantifying the contributions of each individual feature. Figure 2 shows the feature classification from the most important to the less important, where Class 2 represents severe pain, Class 1 represents moderate pain, and Class 0 represents no pain status.

## 5. Discussion

This study demonstrates the feasibility of combining physiologic data collected on wearable devices with machine learning techniques to predict pain scores in an elderly population with neurodegenerative diseases and cognitive impairment. In addition, our study shows that in the selected population, it is not necessary to use a pain stimulus if the concomitant numerical pain assessment is timely and correctly conducted. For our data analysis, we aimed to build predictive models for pain based on physiologic wearable sensor data labelled according to the pain annotation carried out by the research nurses. The data were obtained with minimal to no risk to patients. Although the number of patients was reduced, and only a limited number of patients had severe pain, the dataset was still diverse and voluminous for a deeper analysis.

The best accuracy was found using the machine learning technique named light gradient-boosting machine, with an accuracy of 0.7047 for the prediction of pain on a three-point scale. Based on the set of used signals, a clear distinction was observed in terms of feature importance between the EDA, temperature, and mobility data, and the second set composed from the IBI, HR, and BVP.

The feasibility results suggest a number of potential approaches to predict pain and perceive pain’s physiological responses. Similar results have been obtained in different configurations, but not for patients with dementia. In a review of several studies, the reporting of promising correlations between pain scores and signal measurements of physical activity and functional outcomes as measured by a wearable accelerometer was established [16]. It was established that applying different statistical features extracted from EDA data collected from postoperative patients with a combination of binary classification models and a machine learning algorithm can be used in the detection of pain intensity [17]. In another study, conducted with 23 healthy volunteers being exposed to provoked pain, the results showed that at least three levels of pain can be quantified with good accuracy and physiological evidence from recorded EDA signals in association with machine learning models [18].

There are limitations to this study. First, we used data without distinction between rest time and activity time. As shown in the feature extraction, the mobility and accelerometer data are among the data that represent the highest correlation. A clear distinction should be added during the labelling process. Second, patients may have had underlying medical conditions that could affect the HR, IBI, and BVP, and they were not considered. Third, each patient underwent pain analysis by a ward nurse, which differed by day and by nurse, and could affect the sensitivity and evaluation. Fourth, we had a small sample of patients with severe pain, probably related to the adequate management in the hospital. Furthermore, we were not able to measure associations with pain at rest or related to activity time, the location or type of pain, or potentially related symptoms of the patients, such as anxiety, fatigue, or quality of life, for example.

The primary results provide clinical information about pain in patients with dementia. Although we collected valuable data from wearable devices, labelling the data correctly to enable their usage in machine learning modelling required a better understanding of their related context. Some of our early data were not included in the final data processing owing to the lack of analogue annotations. Although over 60% of the labelled data contained known raw and written contexts, we had problems with the data regarding the nature of the intervention, the precise time of the intervention, and its related outcome in the annotations. The difficulty with such data is that contextual data require a digital transfer, and it is not easy to perform it immediately after the intervention. Moreover, such data are still not fully integrated into the EHR. Another challenge in annotation is related to the definition of the metrics that are required to describe the context. The information we received about these metrics from the ward nurses was insufficient; thus, we needed to define them ourselves. For example, analgesic administration and the time to effect were standardised, as there was no assessment of patients after the intervention to determine the correct time from which a patient experienced less pain.

The secondary result is regarding the feasibility of our approach. Our findings provide a proof of concept of the feasibility of the real-time assessment of pain among patients suffering from dementia with and without communication impairment. However, we have not noticed any differences in signals or data between the two categories of patients.

Although the conclusions from our research are encouraging, there are still numerous challenges in implementing this study in a large clinical trial setting. First, we believe that pain is a subjective sensation. Its perceptions, such as intensity and frequency or even quality of life, may influence how a person reports pain. Our approach did not include such information declared by a patient. Instead, it used pain intensity information from the wearable signal. Second, the analysis excluded other feelings, such as stress, anxiety, and fatigue. For a future fully holistic analysis, it is necessary to consider these as well. Nevertheless, as we may use wearable technology in such a way, it will help us in further developing a real-time pain assessment approach. Despite these limitations, this is an interesting and innovative approach in the field of pain management among populations with no communication abilities, because it can help to provide effective pain relief while also improving a patient’s quality of life.

## 6. Conclusions

In this feasibility pilot study, we aimed to refine a predictive model with a larger and more diverse dataset. We improved the process of analogue annotation by including further details and standardising the collection process. Additionally, we increased our model’s accuracy by including patients’ chronic pain history and by including a more communicative patient as a comparison group.

We have shown that one can use wearable technologies to estimate pain intensity and frequency among people with dementia. This can be useful to improve their quality of life and the lives of family caregivers. We aim to test our approach in a larger clinical trial, where a new portable pain assessment tool will be used. This new approach will integrate existing home healthcare systems and may lead to the real-time remote monitoring of pain in individuals with dementia. Furthermore, it may facilitate a better interaction between family members, healthcare providers, and community-based rehabilitation teams.

We have proved the feasibility of a simple, convenient, and inexpensive pain measurement approach for non-communicative patients. However, we are yet to formally identifying meaningful clinical uses for this non-invasive pain measurement tool. Future research, with a larger sample of patients and the inclusion of a neural network, will be carried out in order to evaluate predictive models, differences in sub-groups, and the efficiency as well as the reliability of the approach.

## Figures and Tables

**Figure 1 sensors-24-06298-f001:**
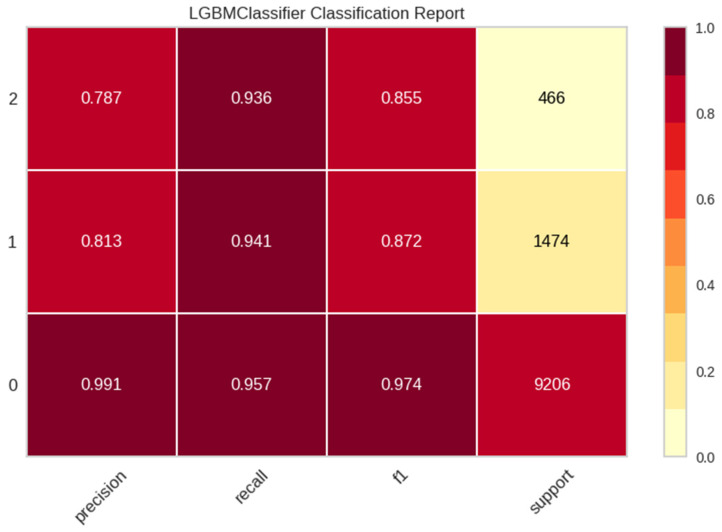
F1 scores and support scores.

**Figure 2 sensors-24-06298-f002:**
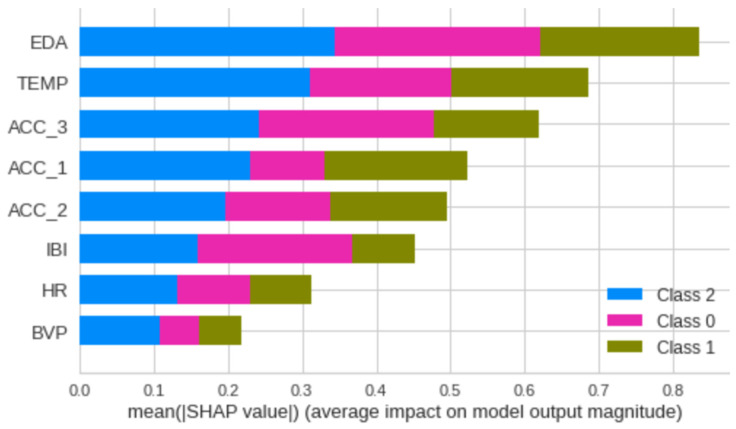
Feature classification.

**Table 1 sensors-24-06298-t001:** Conversion of the pain assessment scores.

	NRS	Algoplus
No pain	0–3	0–1
Moderate pain	4–6	2–3
Severe pain	7–10	4–5

**Table 2 sensors-24-06298-t002:** Demographic and medical characteristics of study participants at baseline.

Age	MMSE x/30	Communication Ability	Comorbidity (Charlson Index)	Pain at Admission	Origin of Pain (1 = Osteoarticular; 2 = Inflammatory; 3 = Neurogenic; 4 = Visceral; 5 = Other)	Other Origin	Daily Analgesic
86	19	Y	4	Y	-	.	Y
85	10	N	5	Y	1	.	Y
85	21	N	7	Y	1	.	Y
89	20	Y	8	Y	1	.	Y
77	0	N	5	N	1	.	Y
87	21	N	6	Y	1	.	Y
91	14	N	6	Y	1, 5	Abdominal	Y
88	17	Y	7	Y	1	.	N
86	17	Y	8	N	4, 5	Headache	Y
86	30	Y	7	Y	1	.	Y
84	23	Y	8	Y	1, 2, 3	.	Y
78	16	Y	8	Y	1, 3	.	Y
79	30	Y	10	Y	1, 2, 3	.	Y
91	26	Y	5	Y	1, 2	.	Y
68	29	Y	9	Y	1	.	Y
89	28	Y	5	Y	1	.	Y
84		Y	13	Y	3, 5	Sores, Ulcers	Y
64	24	Y	17	Y	1, 3, 4	.	Y

**Table 3 sensors-24-06298-t003:** (**a**) List of feature values for moderate pain. (**b**) List of feature values for severe pain.

(a)								
	TEMP	HR	BVP	EDA	IBI	ACC_1	ACC_2	ACC_3
count	37	37	37	37	37	37	37	37
mean	33.36	115.61	0.000052	2.60	0.4820	−8.86	−0.3996	16.59
std	0.15	2.82	0.001659	0.15	0.0121	1.29	2.0729	1.36
min	33.09	109.59	−0.004404	2.32	0.4611	−11.55	−3.8940	13.17
25%	33.28	113.65	−0.000682	2.48	0.4726	−9.60	−1.7539	15.72
50%	33.32	115.20	−0.000193	2.59	0.4821	−8.91	−0.3042	16.54
75%	33.38	117.23	0.000376	2.72	0.4925	−7.86	1.4026	17.52
max	33.75	122.59	0.004505	2.85	0.5054	−6.70	3.1584	19.17
**(b)**								
	**TEMP**	**HR**	**BVP**	**EDA**	**IBI**	**ACC_1**	**ACC_2**	**ACC_3**
count	37	37	37	37	37	37	37	37
mean	30.32	114.40	−0.000017	1.40	0.4778	1.44	0.3996	13.35
std	0.14	3.78	0.000704	0.14	0.0176	3.47	2.0729	2.99
min	30.03	106.20	−0.001795	1.15	0.4508	−5.26	3.8940	7.23
25%	30.20	113.23	−0.000347	1.29	0.4629	−0.71	1.7539	10.94
50%	30.35	114.59	0.000029	1.40	0.4762	1.97	0.3042	13.31
75%	30.45	116.85	0.000331	1.43	0.4903	2.97	1.4026	16.09
max	30.50	120.87	0.001407	1.79	0.5204	10.63	3.1584	17.80

**Table 4 sensors-24-06298-t004:** Summary of the performance of the 14 algorithms.

	Model	Accuracy	AUC	Recall	Prec.
lightgbm	Light gradient-boosting machine	0.655	0.7766	0.6271	0.8111
rf	Random forest classifier	0.804	0.8284	0.6107	0.8239
et	Extra tree classifier	0.8217	0.8316	0.5942	0.8269
dt	Decision tree classifier	0.6909	0.678	0.5669	0.7915
gbc	Gradient-boosting classifier	0.5207	0.691	0.5623	0.7876
knn	K neighbours classifier	0.5905	0.6744	0.5184	0.7726
ada	AdaBoost classifier	0.4266	0.5838	0.4605	0.7424
lda	Linear discriminant analysis	0.4109	0.5823	0.4069	0.7305
lr	Logistic regression	0.4126	0.5821	0.4062	0.7302
ridge	Ridge classifier	0.4097	0	0.4057	0.7306
qda	Quadratic discriminant analysis	0.2007	0.5555	0.3807	0.7443
nb	Naive Bayes	0.1888	0.5518	0.3728	0.7383
svm	SVM—linear kernel	0.387	0	0.3619	0.747
dummy	Dummy classifier	0.8244	0.5	0.3333	0.6796

**Table 5 sensors-24-06298-t005:** Tuned 10-fold cross-validation of the LGBTM classifier.

	Accuracy	AUC	Recall	Prec.
0	0.7023	0.7994	0.6358	0.8103
1	0.704	0.7852	0.6377	0.818
2	0.7123	0.7861	0.6215	0.8088
3	0.6888	0.7603	0.6241	0.7987
4	0.6988	0.7671	0.5914	0.8004
5	0.7075	0.7793	0.6027	0.8068
6	0.7184	0.7761	0.5938	0.8016
7	0.7091	0.7845	0.5954	0.8133
8	0.7057	0.7738	0.6092	0.8056
9	0.7	0.7812	0.6216	0.8036
Mean	0.7047	0.7793	0.6133	0.8067
SD	0.0077	0.0103	0.0163	0.0057

## Data Availability

The original contributions presented in the study are included in the article, further inquiries can be directed to the corresponding author.

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
