# Peer review of "Pain Assessment for Patients with Dementia and Communication Impairment: Feasibility Study of the Usage of Artificial Intelligence-Enabled Wearables"

_sensors, 2024, doi:10.3390/s24196298_

Round 1

Reviewer 1 Report

Comments and Suggestions for Authors

The manuscript entitled "Pain Assessment for Patients with Dementia and 2 Communication Impairment: Feasibility Study of the Usage of 3 Artificial Intelligence-Enabled Wearables" could be potential study to highlight the involvement of artificial intelligence-enabled wearables in pain assessment. However, the sample size incorporated in the study is very low, and hence the authors arte required to increase the sample size. Further, the authors should distinguish which kind of pain and area of the pain in the study. Additionally, the authors used Support Vector Machine (SVM), Logistic Regression, Decision Tree, and Random Forest models. It is quite good if the authors try to use neural networks for their study or try to make a comparitive chart with all used models and neural networks. 

Comments on the Quality of English Language

There are some typos and sentence errors, and henec authors are suggested to check the entire manuscript very cautiously.

Reviewer 2 Report

Comments and Suggestions for Authors

1.     The article's content is a feasibility study regarding the use of artificial intelligence-enabled wearable devices for pain assessment in elderly patients with dementia and communication impairment.The title pertains to pain assessment for patients with dementia and communication impairment.Although dementia is usually related to increasing age and is more prevalent in older populations, particularly those above 65 years old, this does not imply that dementia only affects the elderly. Moreover, the text repeatedly mentions that the perception of pain is more subjective and cognitively based. The study did not consider the subjectivity and frequency of pain, as well as their correlation with quality of life and other related information. Therefore, is there a need for empirical evidence from papers or experimental data to support the differences in pain among patients of various age groups? At the same time, no differences in signals or data were observed between patients who could communicate and those who could not, suggesting that more data may be required to distinguish between these groups.

2.     In the article, the collected samples of pain levels (no pain, moderate pain, severe pain) are 7975, 1444, and 442, respectively. The collected data are highly imbalanced, which may affect the accuracy of the predictive model.

3.     The data collected by the sensors in the study primarily targeted signals such as blood pressure, heart rate, skin conductance activity, and interbeat intervals to indirectly reflect the pain assessment of the samples. However, there are many factors that can affect these physiological signals, such as the patient's stress, anxiety, and fatigue or others. At the same time, the study did not differentiate the data between rest time and activity time, as well as potential influences like medical equipment, which could affect the interpretation of the data, as the level of activity may affect physiological measurements.

4.     In the study, some technical issues in the data collection and processing were mentioned, such as limitations in the device's battery life and storage capacity. However, the study did not demonstrate whether the device could ensure the signal-to-noise ratio and other performance metrics under different battery levels and storage capacities.

Comments on the Quality of English Language

No

Reviewer 3 Report

Comments and Suggestions for Authors

Thank you for this submission. A very important and timely piece of research which has the potential to improve the lives of older adults with dementia who may be experiencing pain. A potential aid to assessment and therefore management of pain. A worthwhile feasibility study. I cannot see any reference to ethical approval, can the authors please confirm that ethical approval has been obtained and add the reference number.

Round 2

Reviewer 1 Report

Comments and Suggestions for Authors

Manuscript is accepted in its current form

Reviewer 2 Report

Comments and Suggestions for Authors

Accept